# Revolution and Control of Fe-Al-(Mg, Ti)-O Oxide Inclusions in IF Steel during 260t BOF-RH-CC Process

**Rijin Cheng** [1,2], **Renchun Li** [2], **Di Cheng** [2], **Junshan Liu** [2], **Qing Fang** [1], **Jian'an Zhou** [1], **Wenliang Dong** [3], **Hua Zhang** [1,*] and **Hongwei Ni** [1,*]

[1] The State Key Laboratory of Refractories and Metallurgy, Wuhan University of Science and Technology, Wuhan 430081, China; chengrijin@wust.edu.cn (R.C.); qingfang@wust.edu.cn (Q.F.); zhou_jianan@sina.com (J.Z.)
[2] Technical Center, HBIS Group Hansteel Company, Handan 056001, China; lirenchun@hbisco.com (R.L.); Chengdi@hbisco.com (D.C.); liujunshan@hbisco.com (J.L.)
[3] Research Institute of Technology, Shougang Group Co., Ltd., Beijing 100043, China; dongwenliang002@163.com
* Correspondence: huazhang@wust.edu.cn (H.Z.); nihongwei@wust.edu.cn (H.N.); Tel.: +86-27-68862811 (H.Z. & H.N.)

**Abstract:** The evolution of inclusions that contain Al, Mg, and Ti was studied through industrial-grade experiments. Field emission scanning electron microscopy, energy dispersive spectrometry, inductively coupled plasma atomic emission spectrometry, and FactSage software were used to analyze the evolution mechanisms of inclusions in Al-killed titanium alloyed interstitial free (IF) steel. The research found that the evolution of inclusions during the smelting process of IF steel is results in 'large sphere-like $SiO_2$-CaO-FeO-MgO-MnO' and 'small cluster spherical FeO-MnO' change to cluster-like $Al_2O_3$ and irregular $MgO \cdot Al_2O_3$, then change to $Al_2O_3 \cdot TiO_x$ and $Al_2O_3$, and finally change to $Al_2O_3$. It is difficult for $Al_2O_3 \cdot TiO_x$ to stably exist in the IF molten steel. It is the key to extend the holding time properly after Ruhrstahl Heraeus (RH) to ensure the removal of $Al_2O_3$ inclusion. With the increase of Mg content, the change path of $MgAl_2O_4$ inclusion in IF steel is that $Al_2O_3$ changes to $MgO \cdot Al_2O_3$, and finally changes to MgO. It is difficult to suppress $MgO \cdot Al_2O_3$ spinel formation by controlling the oxygen in the steel, but Ca can modify part of the $MgO \cdot Al_2O_3$ spinel inclusions during RH refining. In order to ensure the removal of 6–10 μm inclusions, the holding time is suitable for 19–42 min.

**Keywords:** IF steel; inclusion evolution; thermodynamics; $MgAl_2O_4$ inclusion; $Al_2O_3 \cdot TiO_x$

## 1. Introduction

Ultra-low carbon interstitial-free (IF) steel is widely used in automobile plate production because of its excellent deep draw ability and uniform mechanical properties. With the increasing quality requirements of cold rolling sheet, the control requirements of inclusions in ultra-low carbon steel are increasingly strict [1,2]. In the smelting process of ultra-low carbon IF steel, a certain amount of titanium, niobium, and other elements should be added, and the interstitial atoms, such as carbon and nitrogen, in ultra-low carbon steel should be completely fixed as carbon nitrogen compounds, so as to obtain clean ferritic steel without interstitial atoms [3]. According to the difference in the chemical composition of the inclusions, the inclusions in ultra-low carbon steel can be divided into $Al_2O_3$ inclusions, $Al_2O_3$-$TiO_x$ inclusions, and CaO-$Al_2O_3$-MgO inclusions [4–7]. Such inclusions can easily block immersion nozzles during continuous casting, and can also cause defects in the product. The study shows that $Al_2O_3$-$TiO_x$ inclusions are closely related to the clogging of the nozzle and the final product defect [8].

$Al_2O_3$-$TiO_x$ inclusions are easily formed when the content of FeO is high [4], and the inclusions are unstable [6]. The [Al] and [Ti] react with the [O] from the self-dissociation of $SiO_2$ in the slag and form inclusions changing from solid $Al_2O_3$ to $Al_2O_3$-$TiO_x$ inclusions. The inclusion chemistry composition is mainly dependent on the $w([Al])$ and $w([Ti])$, while the number of inclusions can be reduced by increasing the ratios of CaO to $SiO_2$ and CaO to $Al_2O_3$ in the slag [9]. Rare earth Ce is usually used to modify spinel inclusions in steel [10]. The irregular $Al_2O_3$ inclusions will be modified by the rare earth element Ce in Al-killed titanium-alloyed IF steel. The irregular $Al_2O_3$ inclusions of 10–15 μm are wrapped by rare earth, and gradually transformed into spherical $CeAlO_3$, $Ce_2O_3$, and $Ce_2O_2S$ inclusions of 5 μm, and finally dispersed into IF slabs [11].

When T[O] in steel is reduced to 0.001%, the spinel inclusions in molten steel become the most important factor affecting the purity of steel [12]. The $MgO·Al_2O_3$ spinel will be modified by Ca treatment. The inclusion modification route is $Al_2O_3$ change to $MgO·Al_2O_3$, and finally change to liquid complex inclusions [13]. Mg migrates to the molten steel and $MgAl_2O_4$ spinel inclusion is formed due to a reaction between Mg and $Al_2O_3$ inclusions. The spinel inclusion changes entirely into liquid oxide inclusion via the transfer of Ca from slag to metal. The modification reaction is more efficient as the $SiO_2$ content in the slag decreases [14]. When the content of Al in steel is constant, with the increase of Mg content in steel, the inclusions in steel continuously precipitate $MgAl_2O_4$ spinel and finally changes into liquid MgO [15]. It is very important to control the content of Mg and Al in the alloy and to prevent the secondary oxidation of molten steel [16].

In summary, the inclusions have an important impact on the blockage of immersion nozzles and product quality of ultra-low carbon steel. In actual production, the molten steel in the basic oxygen furnace (BOF)-Ruhrstahl Heraeus (RH)-continuous casting (CC) process was sampled, and the characteristics, formation and evolution process, formation mechanism, and influencing factors of the inclusions were analyzed in detail to provide a theoretical basis for the control of inclusions in ultra-low carbon steel. Finally, the mechanical properties of DC06 IF steel were tested, and it was confirmed that the improved IF steel met the requirements of relevant steel grades. The experimental results have a complete understanding of the evolution process of inclusions in IF steel, provide guidance for the evolution and control of $MgO·Al_2O_3$ spinel inclusions and $MgO·TiO_x$ inclusions in steel, and put forward the process parameters to reduce the number and size of inclusions in steel. This is of great significance for improving the control level of non-metallic inclusions in automobile plate steel.

## 2. Materials and Methods

### 2.1. Materials

DC06 IF steel was produced by the 260 t BOF-RH-Holding-CC process at the Hanbao steel plant. During tapping, 600 kg lime was added into the BOF, and 680 kg aluminum slag deoxidant was added onto the top slag after tapping. RH vacuum treatment includes decarburization, followed by aluminum deoxidization, then 338 kg Ti-Fe containing 70% Ti alloying and keeping RH pure circulation time for 8–10 min. After RH treatment, molten steel remained unstirred in the ladle for 25–45 min before casting. The standard for judging the chemical composition of DC06 IF steel is shown in Table 1.

**Table 1.** The standard for judging the chemical composition of IF steel, wt%.

| C | Si | Mn | P | S | $Al_s$ | Ti | B | N |
|---|---|---|---|---|---|---|---|---|
| ≤0.0020 | ≤0.010 | 0.08–0.14 | ≤0.012 | ≤0.009 | 0.020–0.050 | 0.065–0.075 | 0.0003–0.0008 | ≤0.003 |

In order to investigate the evolution of inclusions in the IF steel smelting process, samples were taken by samplers of Φ 30 mm × 10 mm during the industrial experiment steelmaking process, and the whole industrial experiments were carried out over 6 heats in total. A schematic diagram of charging and taking specimens during the steelmaking processes is shown in Figure 1.

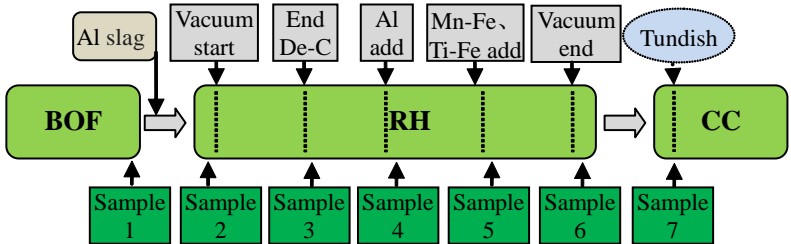

**Figure 1.** Schematic of charging and taking specimens during steelmaking processes.

## 2.2. Mechanical Property Experiment

During the tensile experiment, the DC06 IF steel coil transverse sample was taken, and the steel plate was processed into a dumbbell-shaped sample by using the Zwick 2Z50 (Ennepeta, Nordrhein-Westfalen, Germany) sample preparation machine. The initial gauge lengths of the tensile samples $l_0$ and $b_0$ are 80 mm and 20 mm, respectively. Then the XKA714C (BYJC, Beijing, China) automatic numerical control machine was used to mill the edge of the sample to make it meet the standard GB/T228.1-2010. According to the inspection standard GB/T5213-2019, the Zwick automatic tensile testing machine (Model Z150robo Test L, Ennepeta, Nordrhein-Westfalen, Germany) was used to test the tensile test samples to obtain mechanical performance data, whereby 700 sets of tensile tests were carried out on DC06 IF steel, and the average value of mechanical properties was obtained.

## 2.3. Methods of Chemical Analysis

Each steel sample on the cross-sectional had been ground and polished by SiC papers and w1.5 diamond suspensions. The sample should be ground and polished in the way of transverse and longitudinal intersections, and inclusions of each steel sample were detected using FEI Nova NanoSEM400 (FEI, Hillsboro, OR, USA) scanning electron microscopy (SEM) coupled with energy-dispersive X-ray spectroscopy (EDS, FEI, Hillsboro, USA). The number, size, and chemical composition of inclusions were analyzed automatically using an ASPEX scanning electron microscope (ASPEX SEM, FEI, Delmont, PA, USA). The concentration of the Al, Ti, and Ca in steel was determined by IRIS advantage radial inductively coupled plasma atomic emission spectrometry (ICP-AES, Thermo Elemental, MA, USA). The [O] concentration in steel was measured online by a MSO-3690oxygen sensor (MINCO, Harland, WI, USA).

## 3. Results and Discussion

### 3.1. Morphology and Evolution of Typical Non-Metallic Inclusions in Molten Steel

Figure 2 shows the morphology and evolution of typical inclusions in molten steel. Sample 1 was taken from BOF before tapping. The morphology of inclusions before tapping is shown in Figure 2a. Inclusions in BOF steel are spherical. The inclusions in the BOF process are $SiO_2$-$CaO$-$FeO$-$MgO$-$MnO$ type multi-phase composite inclusions. Due to the deep decarburization by oxygen blowing in BOF, the molten steel has strong oxidizability. Therefore, there are many cluster spherical $FeO$-$MnO$ inclusions below 5 μm in the molten steel, and other large sphere-like inclusions below 50 μm are $SiO_2$-$CaO$-$FeO$-$MgO$-$MnO$.

Sample 2 was taken from ladle before vacuum start. Figure 2b shows typical inclusions in molten steel before vacuum start during RH refining. The spherical and lump inclusions in the steel further grow up and float up, and many $CaO$-$SiO_2$-$FeO$-$Al_2O_3$-$MgO$-$MnO$ inclusions are in the 50 μm size. Due to the addition of Al slag modifier, large particles of $Al_2O_3$ above 50 μm appear in the steel.

Sample 3 was taken from ladle after decarburization. After decarburization by RH vacuum refining, typical inclusions in molten steel are shown in Figure 2c. Irregular $FeO$-$MnO$-$MgO$ and $MgO$-$Al_2O_3$ inclusions combine to form approximately 15 μm spherical inclusions or irregular $FeO$-$MnO$-$MgO$ and $CaO$-$Al_2O_3$-$FeO$-$SiO_2$ inclusions combine to form approximately 50 μm nearly spherical inclusions.

Large inclusions have been partially removed, and the single $Al_2O_3$ has been combined with other inclusions to form complex inclusions.

Sample 4 was taken from ladle after adding aluminum. Typical inclusions in molten steel after adding aluminum for 2 min are shown in Figure 2d. After decarburization in the RH refining process, aluminum is added to the molten steel for deep deoxidization. Due to the reaction between aluminum and oxygen, a large number of cluster-like or coral-like $Al_2O_3$ inclusions are generated by reaction (1). In addition, the inclusions in the steel aggregate and grow, and the lump $MgO \cdot Al_2O_3$ spinel inclusions are wrapped by compact coral-like $Al_2O_3$ inclusions. Other Ca-based and Si-based inclusions are relatively rare after floating and removal.

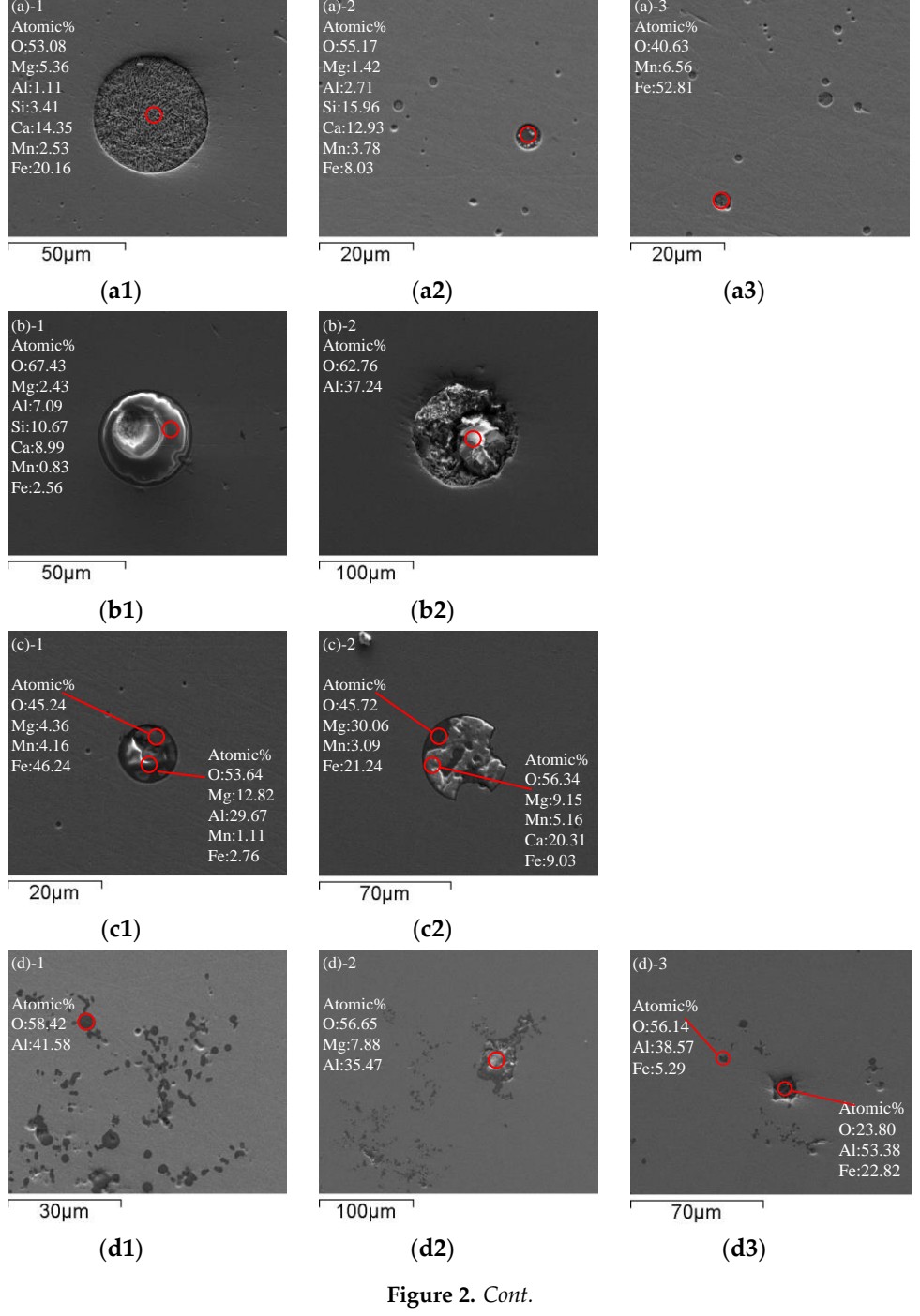

**Figure 2.** *Cont.*

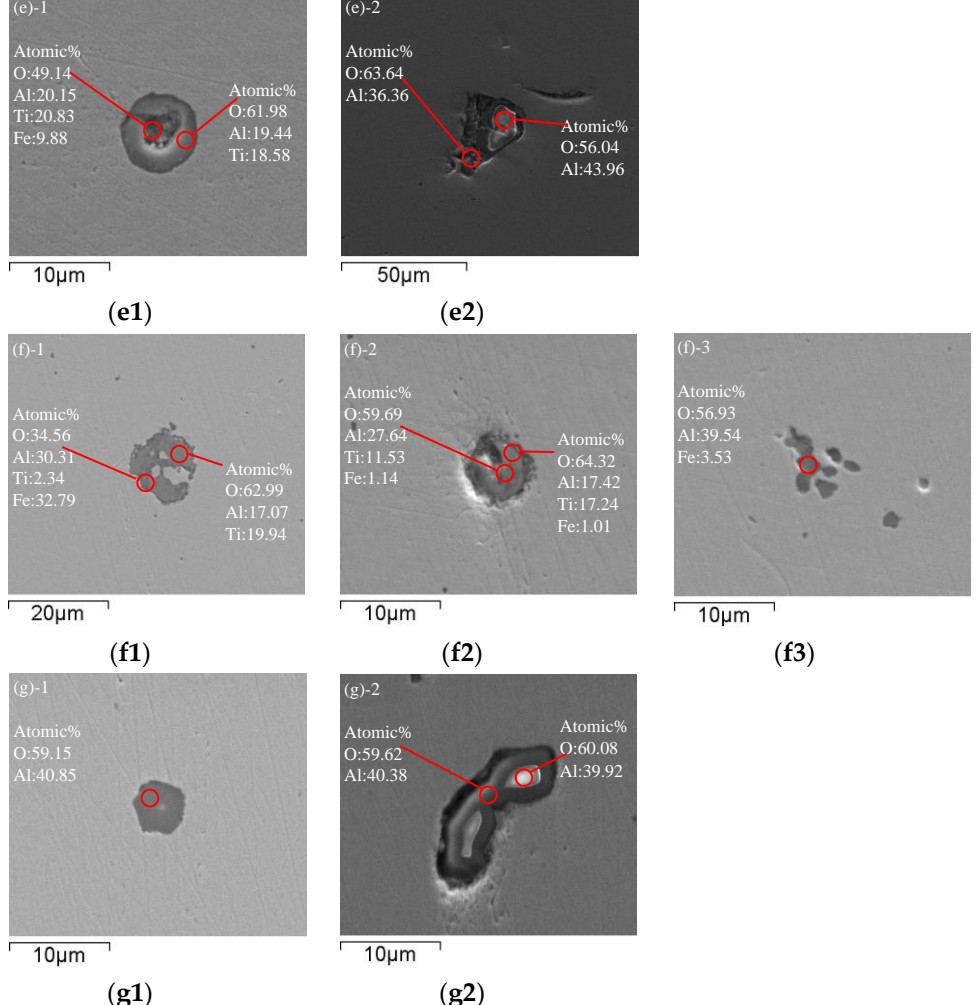

**Figure 2.** Typical inclusions in molten steel: (**a**) Before tapping; (**b**) Before vacuum start; (**c**) After decarburization; (**d**) After adding aluminum for 2 min; (**e**) After adding titanium for 2 min; (**f**) After RH treatment; (**g**) From tundish.

$$[Al] + [O] = (Al_2O_3)_{inclusion} \tag{1}$$

Sample 5 was taken from ladle after adding titanium alloy. Figure 2e shows typical inclusions in molten steel after RH refining and adding titanium alloy. After adding aluminum for deoxidization during RH refining, a large number of cluster-like $Al_2O_3$ inclusions are generated in the steel. With the floating of $Al_2O_3$ inclusions, the total oxygen content in the steel is greatly reduced. Subsequently, after adding Ti-Fe alloying, some [Ti] reacts with [Al] and [O] by reaction (2) in the steel to form $Al_2O_3 \cdot TiO_x$ inclusions below 10 μm, and a part of [Ti] reacts with [O] to generate $TiO_x$, and the $TiO_x$ inclusion wraps around the outside of the $Al_2O_3$ inclusions to produce irregular lump $Al_2O_3 \cdot TiO_x$ inclusions by reaction (3).

$$[Ti] + [Al] + [O] \rightarrow (Al_2O_3 \cdot TiO_X)_{inclusion} \tag{2}$$

$$[Ti] + [O] + (Al_2O_3)_{inclusion} \rightarrow (Al_2O_3 \cdot TiO_X)_{inclusion} \tag{3}$$

Sample 6 was taken from ladle after RH treatment. After RH refining is completed, typical inclusions in molten steel are shown in Figure 2f. At this time, large-scale $Al_2O_3 \cdot TiO_x$ inclusions in the molten steel above 20 μm have floated and removed, but there are still newly generated $Al_2O_3 \cdot TiO_x$

inclusions below 20 μm in the molten steel, and $Al_2O_3 \cdot TiO_x$ grows with $Al_2O_3$ as the core and wraps $Al_2O_3$, and the inclusions are mainly $Al_2O_3 \cdot TiO_x$, except for a few clusters of $Al_2O_3$ in the molten steel.

Sample 7 was taken from tundish during continuous casting. Figure 2g shows typical inclusions in the molten steel of tundish. After the molten steel remains unstirred in the ladle for 25–45 min during the holding process, most of the large-particle $Al_2O_3$ inclusions have been removed by floating. Since the molten steel is sufficiently homogenized, the $Al_2O_3 \cdot TiO_x$ inclusions cannot stably exist in the steel, so the $Al_2O_3 \cdot TiO_x$ inclusions have decomposed into stable $Al_2O_3$ inclusions before entering the continuous casting mold, and Ti re-enters the molten steel. It can be proven from Figure 2g that the inclusions in the final molten steel are mainly $Al_2O_3$. Therefore, it is key to ensure the quality of IF steel to extend the holding time properly after RH to ensure the removal of $Al_2O_3$ inclusions.

### 3.2. Thermodynamics of Evolution and Control of Typical Oxide Inclusions during Refining

The predominance area diagram of Fe-Al-(Mg, Ti)-O system was calculated by using the thermodynamic software FactSage 7.0 (ThermFact Inc., Montreal, QC, Canada). The evolution and control of typical Fe-Al-(Mg, Ti)-O system inclusions in IF steel smelting process are analyzed by using these diagrams.

#### 3.2.1. Typical Spinel Inclusions in Molten Steel before RH Refining

Figure 3 shows the predominance area diagram of the Fe-Al-Mg-O system with different oxygen contents at 1600 °C. The Fe-Al-Mg-O system is used to study the MgO, $MgAl_2O_4$ spinel and $Al_2O_3$ phase diagrams obtained from the change of Al content from $1 \times 10^{-6}$ to 1% and the change of Mg content from $0.1 \times 10^{-6}$ to $100 \times 10^{-6}$. When the Mg content is lower than the critical line of the $Al_2O_3$ and $MgO \cdot Al_2O_3$ phases, an $Al_2O_3$ phase is formed. At this time, the [Mg] content in the molten steel is very low, and it is dissolved in the molten steel in the form of elemental Mg. The Mg content is higher than the boundary line of $MgO \cdot Al_2O_3$ and MgO, and Mg exists as MgO inclusions phase.

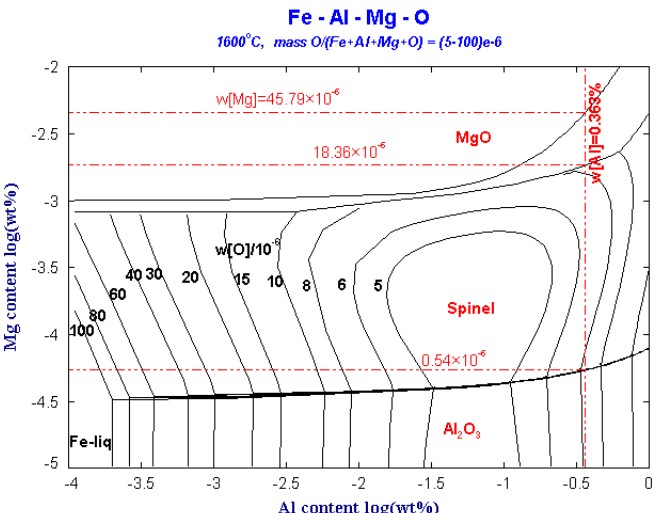

**Figure 3.** Predominance area diagram of Fe-Al-Mg-O system with different oxygen contents at 1600 °C.

The steel samples were taken before RH, after RH, and in the tundish, and the samples were taken from six heats industrial experiment. The chemical compositions of Al, Ti, and Ca elements in the steel are analyzed, and the average compositions are shown in Table 2.

**Table 2.** The average chemical compositions of Al, Ti, and Ca elements in the steel, wt%

| Process | Al | Als | Ca | Ti |
|---|---|---|---|---|
| Before RH | 0.472 | 0.363 | – | – |
| After RH | 0.044 | 0.042 | 0.0001 | 0.077 |
| Ladle holding | 0.036 | 0.033 | 0.0001 | 0.070 |

The ladle slag is modified by Al slag modifier after tapping. Although the Al in the molten steel is about 0.363%, the oxygen in the molten steel is still high. When the content of Mg in the molten steel is $0.54 \times 10^{-6}$, $MgAl_2O_4$ can stably exist in the steel. When the Mg content in the steel is greater than $18.36 \times 10^{-6}$, Mg will coexist as $MgAl_2O_4$ and MgO inclusions. When the Mg content in the steel is larger than $45.79 \times 10^{-6}$, Mg exists only as MgO inclusion. Therefore, as long as the Mg concentration in the molten steel is $(0.54–45.79) \times 10^{-6}$, the $MgAl_2O_4$ spinel inclusions will exist in the molten steel.

3.2.2. Typical Mg-Al Spinel Inclusions in Molten Steel after RH Refining

Figure 4 shows the predominance area diagram of Fe-Al-Mg-Ca-O system with $1 \times 10^{-6}$ Ca and different oxygen contents at 1600 °C. As can be seen from Figure 4, when the content of [Mg] is fixed, the type of inclusions generated in the molten steel is always fixed within a certain range regardless of the change in the [Al] content. Therefore, in the Fe-Al-Mg-O-Ca molten steel system, the type of inclusions is mainly affected by the [Mg] content. In addition, as the [O] content in the molten steel increases, the range of Al corresponding to the formation of $MgAl_2O_4$ spinel and $Al_2O_3$ phases increases. In other words, the probability of forming brittle inclusions such as $MgO \cdot Al_2O_3$ spinel and $Al_2O_3$ increases with increase the [O] content in the steel. However, it is difficult to suppress $MgO \cdot Al_2O_3$ spinel formation by controlling the oxygen in the steel.

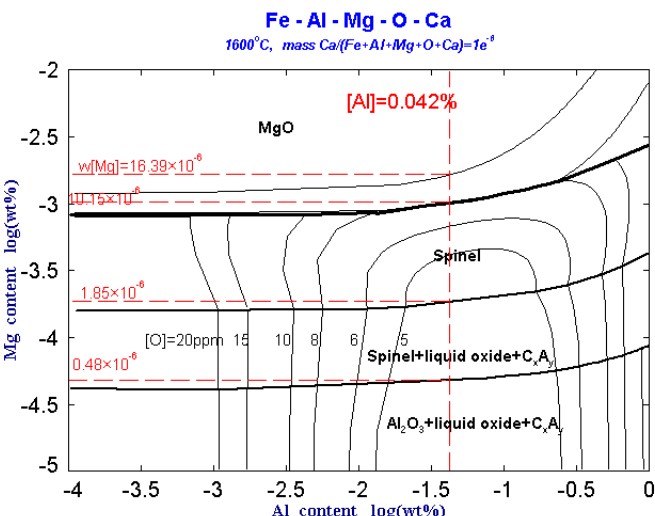

**Figure 4.** Predominance area diagram of Fe-Al-Mg-Ca-O system with $1 \times 10^{-6}$ Ca and different oxygen contents at 1600 °C.

In the Figure 4, 'liquid oxide' represents $CaO-Al_2O_3-MgO$ series liquid inclusions, and '$C_xA_y$' represents $(CaO)_x(Al_2O_3)_y$. When $1 \times 10^{-6}$ [Ca] is added to the molten steel, the $MgO \cdot Al_2O_3$ spinel area is significantly reduced, a part of the $MgO \cdot Al_2O_3$ spinel area is replaced by the 'liquid oxide', and the original $Al_2O_3$ phase area is also replaced by a part of the 'liquid oxide'.

For the IF steel containing 0.042% Al after RH refining in this experiment, when the content of [Mg] in the steel is less than $0.48 \times 10^{-6}$, Mg is dissolved in the molten steel and the inclusions in the molten steel are $Al_2O_3$ and $C_xA_y$ type calcium aluminates. The overall area ratio is larger than that without Ca. When content of [Mg] in steel increases to $0.48 \times 10^{-6}$, $MgO \cdot Al_2O_3$ spinel inclusions begin to generate in the molten steel and it also contains $Al_2O_3$ and $C_xA_y$ type calcium aluminates.

When the content of [Mg] in the molten steel increases to $1.85 \times 10^{-6}$, the Al-containing inclusions in the molten steel all exist as $MgO \cdot Al_2O_3$ spinel. As the content of [Mg] in the molten steel increases to $10.15 \times 10^{-6}$, the inclusions in the molten steel include MgO inclusions besides $MgO \cdot Al_2O_3$ spinel. When the content of [Mg] in the molten steel increases to $16.39 \times 10^{-6}$, the Mg-containing inclusions in the molten steel all exist as MgO.

### 3.2.3. Typical Al-Ti Inclusions in Molten Steel after RH Refining

Figure 5 shows the predominance area diagram of Fe-Al-Ti-O system with different oxygen contents at 1600 °C. It can be seen from the figure that $Al_2O_3$ can remain stable in the molten steel. When the content of [O] in steel is less than $10 \times 10^{-6}$, in addition to the [Al] and [Ti] dissolved in the molten steel, other trace Al and Ti in the steel are stable in the form of $Al_2O_3$ and $Ti_3O_5$, and $Al_2O_3 \cdot TiO_x$ will not be stable in the molten steel.

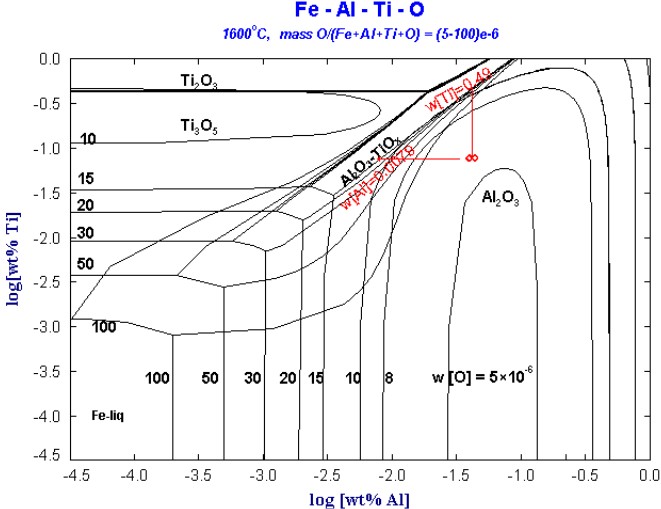

**Figure 5.** Predominance area diagram of Fe-Al-Ti-O system with different oxygen contents at 1600 °C.

When the [O] content in steel increases to $10 \times 10^{-6}$, the stable region of $Al_2O_3$ increases. However, Ti exists as $Ti_2O_3$ instead of $Ti_3O_5$. As the dissolved [O] in steel increases above $15 \times 10^{-6}$, $Al_2O_3$ may react with Ti or $TiO_x$ by reaction (3) or (4) to form $Al_2O_3 \cdot TiO_x$.

$$[Ti] + (Al_2O_3)_{inclusion} \rightarrow (Al_2O_3 \cdot TiO_x)_{inclusion} + [Al] \tag{4}$$

It can be seen from the phase diagram that $Al_2O_3 \cdot TiO_x$ may exist in steel in three situations: (1) when the dissolved [O] in the molten steel is up to $15 \times 10^{-6}$. (2) When the local Ti concentration in the molten steel is up to 0.49% during the titanium alloying adjusting process. (3) When the local Al concentration in the molten steel is as low as 0.0079%. As the dissolved oxygen in ultra-low carbon steel is low, the Al-Ti inclusions in the ultra-low carbon steel liquid cannot exist stably. When the composition of Al and Ti in the steel liquid is uniform, the Al-Ti inclusions will react with the Al in the steel continuously to form $Al_2O_3$ inclusions.

### 3.2.4. Typical Al-Mg Inclusions in Tundish Molten Steel

The average [O] content in the six experimental heats steel dropped to $3.9 \times 10^{-6}$ before Ti adding, but the [O] content increases to $10 \times 10^{-6}$ after RH refining. In order to understand the transformation process of $MgO \cdot Al_2O_3$ spinel, the predominance area diagram of inclusions in Fe-Al-Mg-O system with [O] = 10 ppm and [Ca] = 1 ppm at 1600 °C is shown in Figure 6.

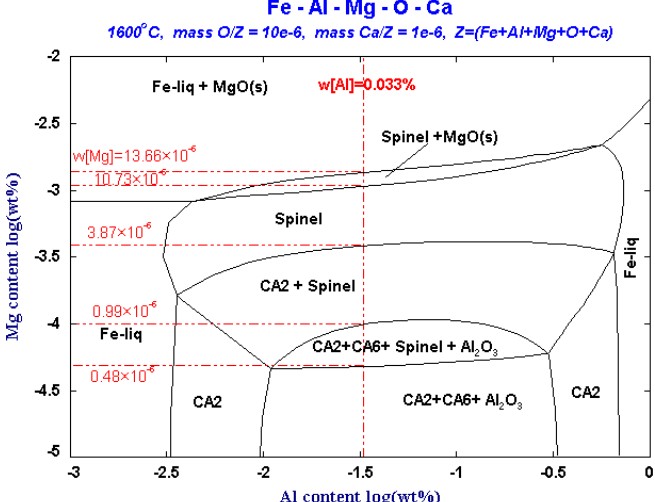

**Figure 6.** Phase diagram of inclusions in Fe-Al-Mg-O system with [O] = 10 ppm and [Ca] = 1 ppm at 1600 °C.

As can be seen from Figure 6, in the Fe-Al-Mg-O system, when the content of [O] and [Al]$_s$ in the steel is $10 \times 10^{-6}$ and 0.033%, respectively, MgO and MgO·Al$_2$O$_3$ and Al$_2$O$_3$ inclusions will be precipitated separately with the difference in Mg content. If [Mg] content is less than $0.48 \times 10^{-6}$ in the steel, Al$_2$O$_3$, CA$_2$, and CA$_6$ inclusions will be precipitated in the molten steel. If [Mg] content increases to $0.48 \times 10^{-6}$, that is, the critical line of the Al$_2$O$_3$ and MgO·Al$_2$O$_3$ phases, MgO·Al$_2$O$_3$ spinel begins to precipitate in the molten steel. At this time, [Mg] and Al$_2$O$_3$ inclusions in the molten steel react to form MgO·Al$_2$O$_3$ spinel at the boundary between the Al$_2$O$_3$ phase and the MgO·Al$_2$O$_3$ phase. As the Mg content increase to $0.99 \times 10^{-6}$, Al$_2$O$_3$ will be transformed to MgO·Al$_2$O$_3$, and the CA$_6$ phase will disappear. When [Mg] content reaches to $3.87 \times 10^{-6}$, Al$_2$O$_3$ will be completely transformed into MgO·Al$_2$O$_3$ spinel, and the CA$_2$ phase will disappear. When [Mg] content continues to increase to $10.73 \times 10^{-6}$, MgO begins to precipitate in the molten steel. When the content Mg increases to $13.66 \times 10^{-6}$, the spinel is completely transformed to MgO. With the increase of Mg content, the change path of inclusions is that Al$_2$O$_3$ change to MgO·Al$_2$O$_3$, and finally change to MgO.

### 3.3. Change and Control of Inclusions during Refining and Holding Time

After the RH deoxidization is completed, the histogram of the change in the number of inclusions in the molten steel at different times is shown in Figure 7. The number, size, and chemical composition of inclusions were analyzed automatically by using ASPEX SEM. As can be seen from Figure 7, the number of inclusions reaches a maximum after 4 min of aluminum deoxidization, which indicates that within 4 min, the oxygen in aluminum and steel reacts quickly to form Al$_2$O$_3$ inclusions. When aluminum is added for 4–8 min, the removal rate of inclusions floating up is greater than the rate of generation, and the total number of inclusions in molten steel decreases, especially the number of inclusions larger than 10 μm. Therefore, in order to ensure that the inclusions generated by deoxidization are fully floated and removed, the pure circulation time after alloying should be greater than 8 min.

After RH refining, the change of the number of inclusions in the molten steel with holding time is shown in Figure 8. It can be seen that the number of inclusions in the molten steel decreases with the extension of the holding time within 20 min, especially the number of large particle inclusions above 10 μm decreases to 1 mm$^{-2}$, and Al$_2$O$_3$·TiO$_x$ also decreases to 1 mm$^{-2}$.

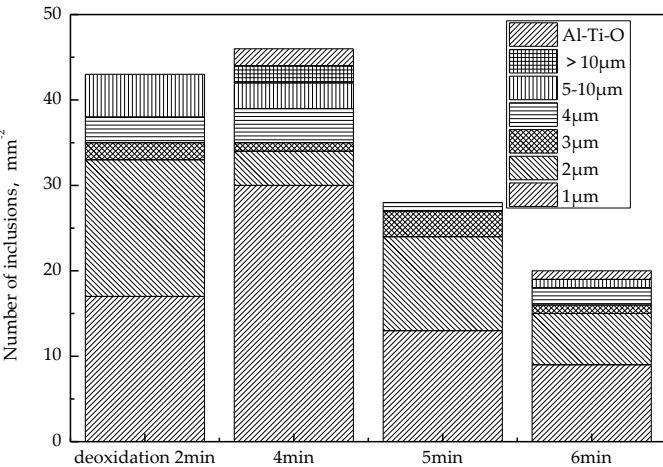

**Figure 7.** Change of inclusions after RH deoxidization.

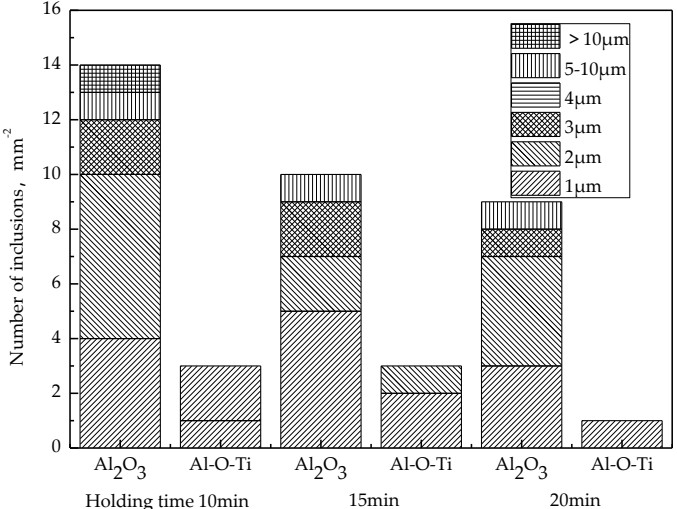

**Figure 8.** Change of inclusions during holding time.

Inclusions float in the molten steel in the way of Stokes to the surface of the ladle. If the time required to reach the top of the molten steel is less than the average residence time of the molten steel, the inclusions can be floated to the top of the molten steel and removed from the molten steel [17].

The floating velocity of inclusions in the still molten steel can be calculated using the Stokes settlement Formula (5).

$$V_s = gd^2(\rho_1 - \rho_2)/(18\eta) \tag{5}$$

where $V_s$ is the floating velocity of the inclusions, cm/s; g is the acceleration of gravity, m/s$^2$; $d$ is the diameter of the inclusions, mm; $\rho_1$ is the density of the molten steel, $\rho_1 = 7.0$ g/cm$^3$; $\rho_2$ is the density of the inclusions, $\rho_2 = 3.5$ g/cm$^3$; $\eta$ is molten steel viscosity, $\eta = 0.05$ g/cm·s at 1600 °C.

It can be seen from the above formula that the floating speed of inclusions is proportional to the square of the diameter of the inclusions, so large inclusions are easy to float. Substituting the data gives:

$$V_s = 0.00381 \times d^2 \tag{6}$$

The depth of molten steel in this ladle is about 3.5 m, and the floating time of inclusions can be calculated by Equation (7), and the result is shown in Figure 9.

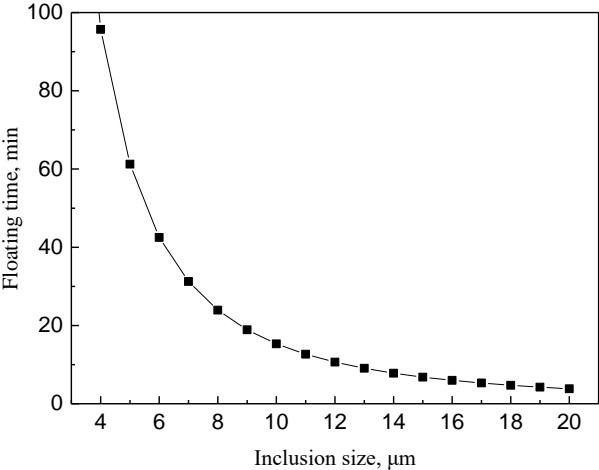

**Figure 9.** The relationship between the particle size and the time of inclusion floating up.

$$t = 350/(0.00381 \times d^2) \tag{7}$$

It can be seen from Figure 9 that inclusions with particle size less than 6 μm need about 42 min to float up. When smaller inclusions aggregate into inclusions with a particle size greater than 9 μm, they can float up and remove within about 19 min, so the holding time should be 19–42 min.

### 3.4. Mechanical Properties of IF Steel

The results of the DC06 IF steel mechanical properties test and requirements (GB/T 5213-2019) are shown in Table 3. According to Table 3, the yield strength $R_{p0.2}$, tensile strength $R_m$, elongation after fracture $A_{80}$, tensile strain hardening index $N_{90}$, and plastic strain ratio $R_{90}$ of DC06 IF steel are 138.7 MPa, 304.7 MPa, 45.3%, 0.24, and 2.91, respectively. Compared with standard values of GB/T5213-2019, the mechanical properties of DC06 IF steel meet the requirements.

**Table 3.** Mechanical properties of DC06 IF steel.

| Standard and Experiment | Yield Strength $R_{p0.2}$, MPa | Tensile Strength $R_m$, MPa | Elongation after Fracture $A_{80}$, % | Tensile Strain Hardening Index $N_{90}$ | Plastic Strain Ratio $R_{90}$ |
|---|---|---|---|---|---|
| GB/T 5213-2019 | ≤170 | 260–330 | ≥42 | ≥0.22 | ≥2.1 |
| Experimental data | 138.7 | 304.7 | 45.3 | 0.24 | 2.91 |

### 4. Conclusions

1. The IF steel inclusions in the BOF process are large sphere-like $SiO_2$-CaO-FeO-MgO-MnO multi-phase composite inclusions below 50 μm and cluster spherical FeO-MnO inclusions below 5 μm. With the addition of Al-slag modifier and Al deoxidizer, a large amount of cluster-like or coral-like $Al_2O_3$ inclusions are formed. Moreover, $MgO·Al_2O_3$ spinel inclusions reduce gradually. $Al_2O_3·TiO_x$ inclusions begin to form after Ti addition, and $Al_2O_3$-type inclusions increase slightly. During the holding process, the inclusions are removed and the number of $Al_2O_3$ inclusions and $Al_2O_3·TiO_x$ inclusions in the steel is greatly reduced. In addition, $Al_2O_3·TiO_x$ inclusions are larger in size compared to $Al_2O_3$ inclusions. After the molten steel remains unstirred in the ladle during the holding process, most of the large-particle $Al_2O_3$ inclusions have been removed by floating. In the tundish, the inclusions of $Al_2O_3·TiO_x$ in the molten steel disappear, and only $Al_2O_3$ inclusions remain in the end. It is key to extend the holding time properly after RH to ensure the removal of $Al_2O_3$ inclusion.

2. The predominance area diagram of Fe-Al-Mg-O system show that, when the Mg concentration in the molten steel before RH is $(0.54–45.79) \times 10^{-6}$, the $MgAl_2O_4$ spinel inclusions will exist in the molten steel. The type of inclusions is mainly affected by the [Mg] content, and it is difficult to

suppress MgO·Al$_2$O$_3$ spinel formation by controlling the oxygen in the steel. When Ca content in steel is $1 \times 10^{-6}$, Ca can modify part of the MgO·Al$_2$O$_3$ spinel inclusions during RH refining, and the amount of Mg required to form spinel is reduced to $(0.48–16.39) \times 10^{-6}$. With the increase of Mg content, the change path of inclusions in IF steel is that Al$_2$O$_3$ change to MgO·Al$_2$O$_3$, and finally change to MgO.

3. Al$_2$O$_3$·TiO$_x$ will not be stable in the IF molten steel, except for three situations: (1) when the dissolved [O] in the molten steel is up to $15 \times 10^{-6}$. (2) When the local Ti concentration in the molten steel is up to 0.49% during the titanium alloying adjusting process. (3) When the local Al concentration in the molten steel is as low as 0.0079%.

4. In order to ensure the removal of 6–10 μm inclusions, the holding time is suitable for 19–42min.

5. The yield strength R$_{p0.2}$, tensile strength R$_m$, elongation after fracture A80, tensile strain hardening index N$_{90}$, and plastic strain ratio R$_{90}$ of DC06 IF steel are 138.7 MPa, 304.7 MPa, 45.3%, 0.24, and 2.91, respectively. The mechanical properties of DC06 IF steel meet the requirements.

**Author Contributions:** R.C.: project administration, conceptualization, investigation, methodology, data curation, writing—original draft preparation. D.C.: Investigation. Q.F.: writing—review and editing. R.L.: project administration. J.L.: investigation, project administration. J.Z.: formal analysis. W.D.: visualization, drawing. H.Z.: funding acquisition, methodology, writing—review and editing. H.N.: conceptualization, supervision. All authors have read and agreed to the published version of the manuscript.

**Funding:** This research was funded by the financial support provided by the Open Youth Fund of State Key Laboratory of Refractories and Metallurgy, Wuhan University of Science and Technology (Grant No. 2018QN03) and and the National Natural Science Foundation of China (51774217).

**Conflicts of Interest:** The authors declare no conflicts of interest.

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
