# Peer review of "Revolution and Control of Fe-Al-(Mg, Ti)-O Oxide Inclusions in IF Steel during 260t BOF-RH-CC Process"

_metals, doi:10.3390/met10040528_

Round 1
Reviewer 1 Report
Dear Authors,
The paper is well written and can be considered for publication after minor corrections. Please follow the points listed below;
- The paper should be extensively edited by native speaking person. Some examples are below: “The chemical composition of IF steel are shown in Table 1” – Should be:“The chemical composition of IF steel is shown in Table 1”. Generally speaking, all is clear and easy for understand, but before publication it will be good to check it again.
- Table 1 – Please add information about the method of chemical analysis.
- Section 3.3. – Can the Authors provide the methodology of the inclusion analysis – how about data scatter?
- Please add also more information about the mechanical properties in the light of the obtained results? Do they participate in any mechanical changes? Please also provide basic strength results for such material. It is recommended to perform greater literature review in terms of this.
Author Response
Point 1: The paper should be extensively edited by native speaking person. Some examples are below: “The chemical composition of IF steel are shown in Table 1” – Should be: “The chemical composition of IF steel is shown in Table 1”. Generally speaking, all is clear and easy for understand, but before publication it will be good to check it again.
Response 1: Thanks for the valuable comment. According to the reviewer’s suggestion, I have made corresponding amendments. Detailed changes are shown in the highlighted text of the abstract.
The author has reviewed the full text again and will seek the help of "English editing" on the "metals" official website before publication.
Point 2: Table 1 – Please add information about the method of chemical analysis.
Response 2: Thanks for the valuable comment. Table 1 is the standard for judging the chemical composition of IF steel, not the test value of the author during the experiment. The concentration of the Al, Ti and Ca in steel was determined by IRIS Advantage Radial inductively coupled plasma atomic emission spectrometry (ICP-AES). The [O] concentration in steel was measured online by MINCO oxygen sensor.
Point 3: Section 3.3. – Can the Authors provide the methodology of the inclusion analysis – how about data scatter?
Response 3: Thanks for the valuable comment. The number, size and chemical composition of inclusions were analyzed automatically by using ASPEX scanning electron microscope.
Point 4: Please add also more information about the mechanical properties in the light of the obtained results? Do they participate in any mechanical changes? Please also provide basic strength results for such material. It is recommended to perform greater literature review in terms of this.
Response 4: Thanks for the valuable comment. The mechanical properties of DC06IF steel were tested, and it was confirmed that the improved IF steel met the requirements of relevant steel grades. The specific description is in the last paragraph of the abstract and experimental part in section 2.2. The experimental results and analysis are in Section 3.4.
Reviewer 2 Report
The paper is clearly structured and understandable. Background is sufficient and conlusions drawn from the investigations are sound.
I have no objections to publish it in the current form.
Author Response
Reviewer 2 have no objections to publish it in the current form. Thank you for your hard work.
Reviewer 3 Report
Dear Authors,
the article is interesting, focused on journal topics and, in my opinion, deserves to be published. Its deals with a specific metallurgical aspect, not so easy to be found in the state of art.
At the same time, the article has to be refined with the scope to improve its overall quality, especially in terms of scientific soundness. Several specific tips are provided inside the attached file. In general, it can be said:
- the abstract has to be fully reconsidered in the way to simply, clarify and offer a more colloquial character. For example, formulas and abbreviations, commonly, have not be included in the abstract.
- innovation of your investigation is not so clear. You should enlarge the last part of the introduction explain what is totally original and unexpected, far beyond the state of arts.
- a 'Materials and Methods' standard section should be included, enlarging and completed the current 'Experiment' section. In particular, it misses a general description of the experimental procedure (e.g. number of tests, equipment, adopted standards, way to elaborate information)
- doublecheck the full text since there are several inaccuracies. For instance, transformations could be better represented using formulas. Formulas can be better represented using a Math Editor. Without these changes, the article is full of potential misunderstandings.
- doublecheck the format: several aspects seem not representing the journal format.
- improve the number and quality of references. In particular, verify if other authors already published similar researches in METALS

Author Response
Point 1: The abstract has to be fully reconsidered in the way to simply, clarify and offer a more colloquial character. For example, formulas and abbreviations, commonly, have not be included in the abstract.
Response 1: Thanks for the valuable comment. For the abbreviations that first appeared in the abstract, I have redefined IF steel and RH as interstitial free (IF) steel and Ruhrstahl Heraeus, respectively. In addition, the "+" and "→" formula symbols in the abstract have been removed and changed into words. Detailed changes are shown in the highlighted text of the abstract.
Point 2: innovation of your investigation is not so clear. You should enlarge the last part of the introduction explain what is totally original and unexpected, far beyond the state of arts.
Response 2: Thanks for the valuable comment. According to the reviewer's opinion, I have made a specific explanation for the innovation of this article in the last part of the introduction, including “The experimental results have a complete understanding of the evolution process of inclusions in IF steel, provide guidance for the evolution and control of MgO·Al2O3 spinel inclusions and MgO·Al2O3 inclusions in steel, and put forward the process parameters to reduce the number and size of inclusions in steel”. The specific content is shown in the highlighted words in the last paragraph part of the introduction.
Point 3: a 'Materials and Methods' standard section should be included, enlarging and completed the current 'Experiment' section. In particular, it misses a general description of the experimental procedure (e.g. number of tests, equipment, adopted standards, way to elaborate information)
Response 3: Thanks for the valuable comment. The author has supplemented the experimental part and added the mechanical properties experiment, including the number of tests, equipment, adopted standards. The contents of this part are highlighted in the article.
Point 4: Double-check the full text since there are several inaccuracies. For instance, transformations could be better represented using formulas. Formulas can be better represented using a Math Editor. Without these changes, the article is full of potential misunderstandings.
Response 4: Thanks for the valuable comment. The full text has been carefully checked. The errors pointed out in the reviewer's PDF article have been corrected. The author deleted the blank space of the abbreviation of the author's name, supplemented the complete spelling of the abbreviation when the professional noun appeared in the article for the first time, removed the formula appearing in the abstract and other texts, corrected formula (6), etc.
Point 5: doublecheck the format: several aspects seem not representing the journal format.
Response 5: Thanks for the valuable comment. According to the format template of the journal, we carefully checked the format of the manuscript and corrected some errors. Such as: A caption on a single line should be centered. No space for author's initials. Do not use formula in summary and text. The top and bottom border width of the three line table is 1.0 point. No comma between issue and year in reference 16.
Point 6: Improve the number and quality of references. In particular, verify if other authors already published similar researches in METALS
Response 6: Thanks for the valuable comment. The literature related to this topic has been searched again, and author has extracted some new ideas from the relevant literature, and has carried on the summary in the introduction. The introduction and the reference have made the supplement. The contents of this part are highlighted in the article.

Round 2
Reviewer 3 Report
Dear Authors,
thank you for considering my remarks.